# Discovery of a cofactor-independent inhibitor of *Mycobacterium tuberculosis* InhA

Yi Xia[1], Yasheen Zhou[1], David S Carter[1], Matthew B McNeil[2], Wai Choi[1], Jason Halladay[1], Pamela W Berry[1], Weimin Mao[1], Vincent Hernandez[1], Theresa O'Malley[2], Aaron Korkegian[2], Bjorn Sunde[2], Lindsay Flint[2], Lisa K Woolhiser[3], Michael S Scherman[3], Veronica Gruppo[3], Courtney Hastings[3], Gregory T Robertson[3], Thomas R Ioerger[4], Jim Sacchettini[4], Peter J Tonge[5], Anne J Lenaerts[3], Tanya Parish[2], MRK Alley[1]

**New antitubercular agents are needed to combat the spread of multidrug- and extensively drug-resistant strains of *Mycobacterium tuberculosis*. The frontline antitubercular drug isoniazid (INH) targets the mycobacterial enoyl-ACP reductase, InhA. Resistance to INH is predominantly through mutations affecting the prodrug-activating enzyme KatG. Here, we report the identification of the diazaborines as a new class of direct InhA inhibitors. The lead compound, AN12855, exhibited in vitro bactericidal activity against replicating bacteria and was active against several drug-resistant clinical isolates. Biophysical and structural investigations revealed that AN12855 binds to and inhibits the substrate-binding site of InhA in a cofactor-independent manner. AN12855 showed good drug exposure after i.v. and oral delivery, with 53% oral bioavailability. Delivered orally, AN12855 exhibited dose-dependent efficacy in both an acute and chronic murine model of tuberculosis infection that was comparable with INH. Combined, AN12855 is a promising candidate for the development of new antitubercular agents.**

## Introduction

*Mycobacterium tuberculosis*, the causative agent of tuberculosis (TB), is a major public health threat. There were an estimated 10.4 million new cases and 1.8 million deaths from TB in 2015 (WHO, 2016). The treatment and eradication of TB is complicated by the emergence and spread of multidrug (MDR)- and extensively drug–resistant strains of *M. tuberculosis*. Consequently, there is a need for new therapeutics that are active against both drug-susceptible and drug-resistant strains.

Standard chemotherapy for drug-susceptible *M. tuberculosis* follows a 6-month drug regimen: 2-months with four drugs (isoniazid [INH], rifampicin, pyrazinamide, and ethambutol) followed by a 4-month period with INH and rifampicin. The frontline antitubercular drug INH is converted from a prodrug to an active form by the catalase–peroxidase KatG (Zhang et al, 1992; Brossier et al, 2016). The activated compound then binds as an INH–NAD adduct to the NADH pocket of the NADH-dependent enoyl-ACP reductase, InhA (Rawat et al, 2003; Vilchèze et al, 2006; Dias et al, 2007). Inhibition of InhA by INH–NAD results in impaired synthesis of mycolic acids (Rozwarski et al, 1993; Vilchèze et al, 2006). Clinical resistance to INH is primarily due to mutations disrupting KatG function that prevent activation of the INH prodrug (Zhang et al, 1992; Seifert et al, 2015; Brossier et al, 2016). Resistance to INH can also be acquired by mutations in the InhA coding sequence and promoter region (Seifert et al, 2015). The *fabG1inhA* C-15T promoter mutation that up-regulates the expression of InhA is present in approximately 20% of INH-resistant clinical isolates (Vilchèze et al, 2006; Seifert et al, 2015). Given the proven druggability of InhA, attempts have been made to bypass KatG-mediated resistance by identifying direct inhibitors of InhA (Pan & Tonge, 2012). Recent examples include the thiadiazoles (GSK693), 2-(*o*-tolyloxy)-5-hexylphenol (PT70), 4-hydroxy-2-pyridines (NITD-916 and NITD-113), and pyridomycin (Luckner et al, 2010; Hartkoorn et al, 2012; Manjunatha et al, 2015; Martínez-Hoyos et al, 2016). Unlike the INH–NAD adduct that competes with NADH binding to InhA GSK693, PT70 and NITD-916 block access to the InhA substrate-binding site by occupying the fatty acyl substrate-binding pocket in a cofactor-dependent manner (Luckner et al, 2010; Hartkoorn et al, 2014; Manjunatha et al, 2015). Pyridomycin is unique in that it binds within the active site of InhA in a manner that blocks both the NADH cofactor and substrate-binding sites of InhA (Hartkoorn et al, 2014). A promising observation from these studies is the lower frequency of resistance for direct inhibitors of InhA with $1 \times 10^{-8}$ for NITD-916 and GSK625 compared with $1 \times 10^{-5}$ for INH (Manjunatha et al, 2015; Martínez-Hoyos et al, 2016). Further studies are required to determine if differences in the in vitro frequency of resistance correlate with reduced resistance frequency in vivo.

[1]Anacor Pharmaceuticals, Palo Alto, CA, USA   [2]TB Discovery Research, Infectious Disease Research Institute, Seattle, WA, USA   [3]Mycobacteria Research Laboratories, Department of Microbiology, Immunology, and Pathology, Colorado State University, Fort Collins, CO, USA   [4]Texas A&M University, College Station, TX, USA   [5]Institute of Chemical Biology and Drug Discovery, Departments of Chemistry and Radiology, Stony Brook University, Stony Brook, NY, USA

Correspondence: tanya.parish@idri.org

In this study, we describe the identification of a novel diazaborine scaffold that inhibits InhA in *M. tuberculosis*. The lead compound, AN12855, binds to and inhibits InhA with submicromolar affinity through a cofactor-independent mechanism resulting in potent activity against drug-susceptible and drug-resistant strains of *M. tuberculosis*. AN12855 exhibited comparable efficacy to INH in both acute and chronic models of TB infection with a lower potential for resistance development and showed in vitro activity against conventional KatG-mediated INH-resistant *M. tuberculosis*. These results suggest that diazaborines are attractive candidates for the development of new anti-TB drugs.

# Results

### Identification of inhibitors of *M. tuberculosis* InhA

Compound screening against purified *M. tuberculosis* InhA identified three initial hits, AN2918 (6-aryloxy-benzoxaborole), AN3438 (5-aryloxy-benzoxaborole), and AN6534 (7-aryloxy-*N*-sulfonyldiazaborine), which had inhibitory concentration ($IC_{50}$) values against the enzyme of 44, 12, and 79 μM, respectively (Table 1). AN3438 and AN6534 had activity against whole-cell *M. tuberculosis* with $IC_{90}$ of 16 and 36 μM, respectively (Table 1). AN2918 was not active ($IC_{90} >$ 200 μM). To validate InhA as the target of this series, we isolated six

**Table 1.** Profiling of diazaborine and oxaborole inhibitors of *M. tuberculosis* InhA.

| Cpd ID | Structure | MW | InhA inhibition (μM) ($IC_{50}$) | H37Rv (μM) $IC_{90}$ | THP-1 intracellular $IC_{50}$ (μM) | THP-1 intracellular $IC_{90}$ (μM) | THP-1 cytotoxicity $IC_{50}$ (μM) | HepG2 cytotoxicity $IC_{50}$ (μM) |
|---|---|---|---|---|---|---|---|---|
| AN2918 |  | 294.0 | 44 | >200 (n = 2) | — | | — | — |
| AN3438 |  | 294.1 | 12 | 16 ± 1.2 (n = 3) | — | | — | — |
| AN6534 |  | 341.2 | 79 | 36 ± 1.7 (n = 3) | — | | — | — |
| AN12541 |  | 427.2 | 0.40 | 0.11 ± 0.21 (n = 5) | 0.046 ± 0.013 (n = 2) | 0.11 ± 0.01 (n = 2) | >50 (n = 1) | >100 (n = 3) |
| AN12855 |  | 441.2 | 0.030 | 0.090 ± 0.050 (n = 10) | 0.021 ± 0.003 (n = 3) | 0.04 ± 0.01 (n = 3) | >50 (n = 2) | >100 (n = 3) |
| AN12908 |  | 441.2 | 2.3 | 7.0 ± 2.6 (n = 3) | — | | — | — |

MW, molecular weight.

**Table 2.** Profile of diazaborine-resistant mutants.

| Strain | InhA SNP | Solid media MIC (µM) | Liquid media IC$_{90}$ (µM) | | |
|---|---|---|---|---|---|
| | | AN3438 | AN6534 | AN12855 | INH |
| H37Rv-LP | WT | 12.5 | 44 | 0.05 | 0.3 |
| LP-AN3438-RM1 | I16T | 25 | 34 | | 0.4 |
| LP-AN3438-RM2 | P151S | 50 | >200 | | 0.1 |
| LP-AN3438-RM3 | D148G | 50 | >200 | | 0.1 |
| LP-AN3438-RM4 | R195Q | 100 | >200 | | 0.4 |
| LP-AN3438-RM5 | E219A | 100 | 26 | | 0.1 |
| LP-AN3438-RM6 | I202T | 25 | 160 | | 0.2 |
| | | | | | |
| LP-AN12855-RM1 | R195L | | | 3.0 | 0.4 |
| LP-AN12855-RM2 | D148G | | | 0.8 | 0.3 |
| LP-AN12855-RM3 | E219G | | | 4.6 | 0.1 |

*M. tuberculosis* strains with resistance against AN3438; the frequency of resistance for AN3438 was $6.5 \times 10^{-7}$ (Table 2). All isolates demonstrated greater than or equal to twofold resistance to AN3438 as compared with the parental H37Rv strain. Four of the strains were cross-resistant to AN6534 with a greater than or equal to fourfold shift in IC$_{90}$ (Table 2). Whole-genome sequencing of these strains identified single nucleotide polymorphisms (SNPs) in *inhA*; these were I16T, D148G, P151S, R195Q, I202T, and E219A (Table 2). Despite SNPs in *inhA*, all strains were fully susceptible to INH (Table 2). Combined, these results suggest that InhA is the molecular target of the diazaborines and the oxaboroles.

### AN2918 and AN3438 form ternary complexes with InhA and NAD$^+$

To understand the binding mechanism of these boron-containing inhibitors, co-crystal structures of AN2918 and AN3438 with InhA were solved to 2.5 and 2.55 Å, respectively (Fig 1A and B and Table S1). The crystal structures revealed several key features about the interaction between InhA and the boron-containing inhibitors: (i) the binding of both AN2918 and AN3438 to InhA was dependent on the formation of a boron covalent bond with the 2′-OH of NAD$^+$ ribose, resulting in ternary complexes of inhibitors with InhA and NAD$^+$; (ii) the negatively charged tetrahedral boron adducts formed by the inhibitors and NAD$^+$ were stabilized by hydrogen bonds with the catalytic residues Tyr158 and Lys165 (Parikh et al, 1999); (iii) the 6-aryloxy group of AN2918 and the 5-aryloxy group from AN3438 both occupy a deep pocket originally occupied by the hydrocarbon chain of a substrate as observed in the structure of InhA bound to the C16-NAC substrate analog (Rozwarski et al, 1999); (iv) the oxime group from AN3438 forms a hydrogen bond with Glu219 from the deep substrate pocket of InhA. This novel hydrogen bond appears to stabilize the salt bridge interactions between Glu219 and Arg195 and the helix 6 conformation; (v) the crystal structure of AN2918, however, shows a disordered substrate-binding site and helix 6, possibly because of lack of a hydrogen bond with Glu219. These data demonstrate that the oxaborole inhibitors AN2918 and AN3438 occupy the substrate-binding site of InhA in an NAD$^+$–dependent

manner. This overall binding mode is similar to previously reported benzodiazaborine inhibitors of the homologous *Escherichia coli* enoyl-ACP reductase (FabI), suggesting possible merging structure activity relationship trends for the oxaborole and diazaborine series (Baldock et al, 1996).

### Synthesis of diazaborines with improved potency against *M. tuberculosis*

AN2918 (6-aryloxybenzoxaborole) had activity against InhA that did not translate to *M. tuberculosis* activity, whereas a close analog without the para-CF$_3$ group was inactive against the enzyme and *M. tuberculosis* (data not shown), whereas AN3438 (5-aryloxybenzoxaborole) was active against both the enzyme and *M. tuberculosis* (Table 1). Incorporation of the oxime from AN3438 and the para-CF$_3$ from AN2918 into the *N*-sulfonyldiazaborine hit, AN6534, dramatically increased potency 350-fold against *M. tuberculosis* and 190-fold against purified InhA for AN12541 (Table 1). Extension of the *N*-methyl sulfonyl group to *N*-ethyl sulfonyl group lead to compound AN12855, further improving potency against both *M. tuberculosis* and the purified enzyme to 0.09 and 0.03 µM, respectively (Table 1). As the aryloxy substituent was tolerated on both 5$^{th}$ and 6$^{th}$ positions of oxaboroles, AN12908 was made to explore optimal substitution site off the diazaborine head. AN12908 had only moderate potency, with an IC$_{90}$ against *M. tuberculosis* of 7.0 µM and IC$_{50}$ against InhA of 2.3 µM (Table 1). We isolated three *M. tuberculosis* isolates resistant to AN12855 at a frequency of $6.5 \times 10^{-7}$. All three isolates contained SNPs in *inhA* that were also identified in isolates resistant to AN3438 confirming on-target activity (Table 2). Mutations in amino acids R195 and E219 resulted in higher resistance levels than mutations in D148G (Table 2). AN12855-resistant isolates retained the WT sensitivity to INH (Table 2). To further investigate InhA as the target of AN12855, we tested the activity of AN12855 against *M. tuberculosis* strains with mutations in *inhA*, the *fabG1inhA* promoter region, or both (Table 3) (McNeil et al, 2017). All of these strains are resistant to the direct

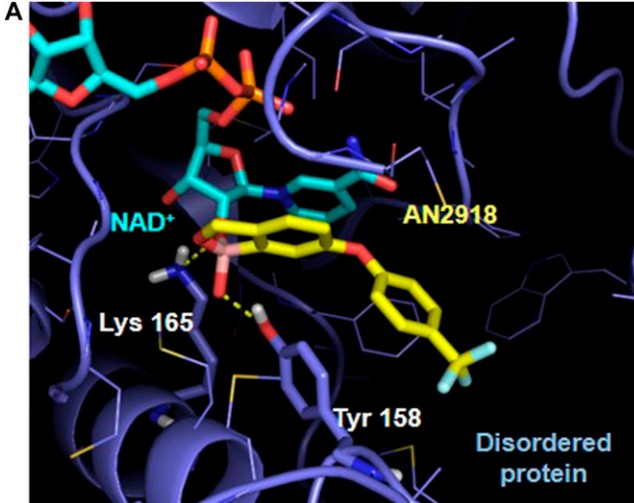

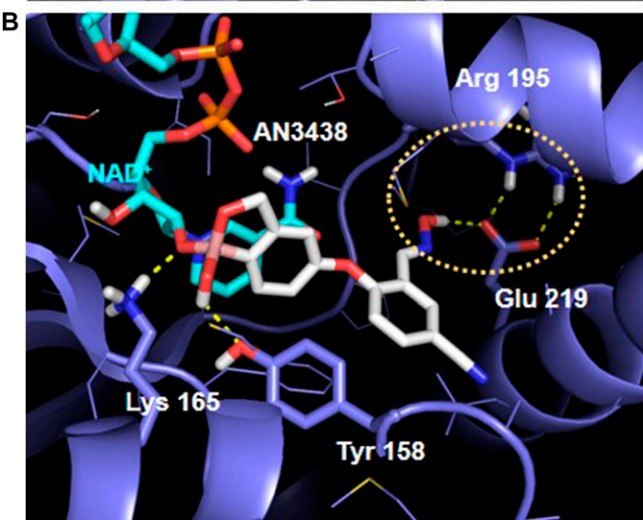

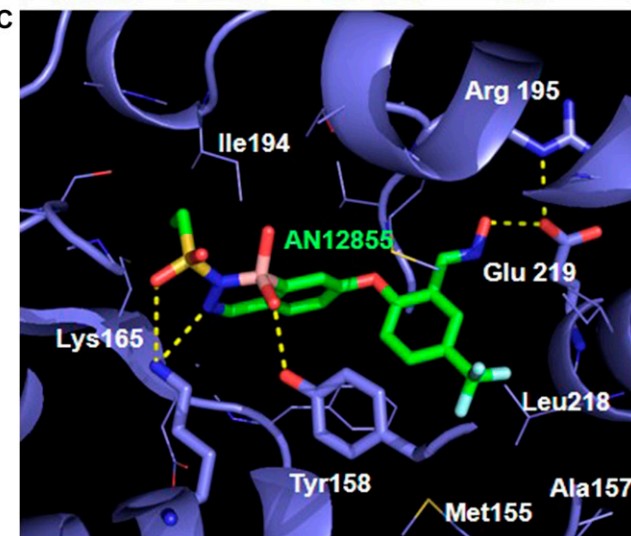

**Figure 1. Complex crystal structures of oxaborole and diazaborine inhibitors with *M. tuberculosis* InhA.**
**(A)** AN2918 (yellow) forms a ternary complex with NAD⁺ (cyan) and InhA (blue). Enzyme residues at the bottom of the deep pocket are disordered. **(B)** AN3438

InhA inhibitor NITD-916 (McNeil et al, 2017). AN12855 was active against most of the strains with mutations in *inhA*, including mutations in I21 and S94 that are observed in INH-resistant clinical isolates (Table 3) (Seifert et al, 2015). Only strains with the D148E, M161L, R195G, or I215S mutations had a greater than fourfold shift in IC$_{90}$ against AN12855 (Table 3). Resistance from InhA$_{D148E}$ and InhA$_{R195G}$ is consistent with the resistant mutants isolated using AN12855 (Table 2). The *fabG1inhA* c-15t promoter mutant strain that overexpresses InhA demonstrated a fivefold increase in IC$_{90}$ against AN12855 (Table 3). Several strains with mutations in both the *inhA* promoter and coding sequence demonstrated high-level resistance, with 10-fold to 85-fold increases in IC$_{90}$ (Table 3). This demonstrates that increased expression of a mutant allele of InhA further increases resistance against AN12855. None of the mutants demonstrated cross-resistance to triclosan (McNeil et al, 2017). Combined, these results support the hypothesis that AN12855 is a direct InhA inhibitor but with a unique binding mode.

In addition, we also tested the activity of AN12855 against three MDR-resistant clinical isolates of *M. tuberculosis*. All strains were INH resistant but were fully susceptible to AN12855 (Table 4). We sequenced *inhA* (including the promoter region) and *katG* in all three strains; strains had mutations or a deletion in *katG* but no mutations in *inhA* (Supplementary Information). In conclusion, AN12855 is a direct InhA inhibitor with potent activity against clinically resistant strains and strains with KatG and InhA coding sequence mutations.

**Inhibition of InhA by AN12855 is cofactor independent**

Isothermal titration calorimetry (ITC) experiments demonstrated that the diazaborine AN12855 was able to bind with InhA in both the presence and absence of NAD⁺ (Fig 2A). The $K_d$ of AN12855 for InhA was 77 nM in both the presence and absence of NAD⁺ (Table 5). AN12855 was unable to bind to InhA in the presence of NADH. Contrasting AN12855, the diazaborine AN12908 with the aryloxy group substituted at the 6ᵗʰ position was dependent on NAD⁺ for binding to InhA (Fig 2B–C and Table 5). Overall, ITC confirmed the increased potency of these inhibitors against InhA and more importantly, revealed that AN12855 binds InhA without a requirement for the cofactor NAD⁺.

To further understand the inhibition mechanism of AN12855, a co-crystal structure of InhA-AN12855 was obtained in the absence of NAD⁺ to 2.65 Å (Fig 1C). Unlike the initial oxaborole hits, this structure revealed a novel binding mode that involved no cofactor and no covalent bonding interaction. Several key features were identified. (i) AN12855 forms a binary complex with InhA and occupies both the cofactor and the substrate sites. (ii) The diazaborine head adopts the negatively charged tetrahedral form interacting with the catalytic residues Tyr158 and Lys165 through hydrogen bonds and charge–charge interactions. (iii) Similar to AN3438, the oxime group of AN12855 forms a hydrogen bond with Glu219 and stabilizes the salt bridge

(gray) forms a ternary complex with NAD⁺ (cyan) and InhA (blue), and a relayed hydrogen-bonding network between the oxime and Glu219 and Arg195 is highlighted. **(C)** AN12855 (green) forms a binary complex with InhA (blue) occupying both the NAD⁺ and the deep substrate pocket. For all images, hydrogen bonds formed between the inhibitors and InhA are highlighted in yellow dash lines. Enzyme residues involved in hydrogen bonds are shown in blue sticks and other key binding residues in blue lines.

**Table 3. Resistance of *M. tuberculosis inhA* promoter and coding sequence mutants against the cofactor-independent diazaborine AN12855.**

| Strain | SNPs[a] | | Liquid $IC_{90}$ (fold shift versus WT)[b] | |
|---|---|---|---|---|
| | *fabG1inhA* promoter (nt) | InhA (Am Ac) | AN12855 | INH[c] |
| H37Rv | — | — | 0.06 µM | 0.1 µM |
| LP-0532543-RM18 | — | S19W | 1× | 1× |
| LP-0532543-RM28 | — | I21M | 1× | 1× |
| LP-0532543-RM34 | — | I21V | 1× | 2× |
| LP-0532543-RM13 | — | F41L | 1× | 1× |
| LP-0532543-RM19 | — | I47L | 3× | 1× |
| LP-0532543-RM1 | — | S94A | 1× | 2× |
| LP-0532543-RM6 | — | M103T | 2× | 1× |
| LP-0532543-RM2 | — | D148E | 5× | 1× |
| LP-0532543-RM4 | — | M161L | 4× | 1× |
| LP-0571426-RM24 | — | I194T | 3× | 3× |
| LP-0532543-RM41 | — | R195G | 16× | 2× |
| LP-0532543-RM9 | — | I202F | 2× | 1× |
| LP-0532543-RM7 | — | G205A | 2× | 1× |
| LP-0532543-RM11 | — | G205S | 2× | 1× |
| LP-0532543-RM3 | — | A206E | 2× | 1× |
| LP-0532543-RM14 | — | G212D | 3× | 1× |
| LP-0532543-RM16 | — | I215S | 15× | 1× |
| LP-0532543-RM37 | — | L269R | 1× | 1× |
| LP-0532543-RM30 | c-15t | — | 5× | 5× |
| LP-0532543-RM314 | c-15t | I47M | 7× | 5× |
| LP-0532543-RM301 | c-15t | N159K | 8× | 3× |
| LP-0532543-RM311 | c-15t | M161V | 84× | 5× |
| LP-0532543-RM304 | c-15t | T162M | 26× | 4× |
| LP-0532543-RM318 | c-15t | M199L | 10× | 4× |
| LP-0532543-RM313 | c-15t | G205D | 13× | 2× |
| LP-0532543-RM320 | c-15t | G208D | 16× | 4× |

[a]No change in sequence from WT H37Rv.
[b]Liquid $IC_{90}$ values WT results are presented as µM, whereas RM IC90 are presented as the fold change compared with WT.
[c]INH values are from McNeil et al (2017).

interactions between Arg195 and Glu219. The large increase in $IC_{90}$ values observed with mutations at Arg195 and Glu219 are consistent with disruptions in the formation of hydrogen bonds between AN12855 and InhA (Tables 2 and 3). (iv) The $CF_3$-substituted aryloxy group fully occupies the deep substrate site tightly packing with hydrophobic residues Tyr158, Ile215, Leu218, Ala157, Met119, and Phe149 (Fig 1C). Resistance to AN12855 in strains with $InhA_{I215S}$ is consistent with disruptions in this substrate-binding site (Table 3). In conclusion, both the ITC and crystal structure of diazaborine AN12855 revealed a novel inhibition mechanism that is independent of the cofactor $NAD^+$.

### Diazaborines are bactericidal for replicating *M. tuberculosis*

We tested two compounds for bactericidal activity under replicating (aerobic growth) conditions. AN12855 demonstrated

concentration-dependent bactericidal activity against replicating *M. tuberculosis* (Fig 3). Under replicating conditions, AN12855 exhibited rapid killing of *M. tuberculosis* at 10× the $IC_{90}$, reaching the limit of detection after 7 d. This rapid bactericidal activity is consistent with targeting of InhA by INH (Fig 3C). Resistance against INH emerged quickly against all concentrations above the $IC_{90}$ (i.e., 0.2 µM) (Fig 3C). Consistent with a reduced resistance frequency, resistance was not observed against the diazaborines (Fig 3A). AN12541 was similarly active under replicating conditions (Fig 3A and B). Thus, inhibition of InhA, with either INH or the diazaborines, is bactericidal against replicating *M. tuberculosis*.

### Diazaborines are active against intracellular *M. tuberculosis*

We tested cytotoxicity for AN12541 and AN12855 against the eukaryotic HepG2 cell line. We observed no cytotoxicity with $IC_{50}$

**Table 4.  Activity of diazaborines against *M. tuberculosis* drug-resistant clinical isolates.**

| Strain | Resistance | MIC (μg/ml) | | | | |
|---|---|---|---|---|---|---|
| | | AN12855 | MOXI | PA-284 | RIF | INH |
| H37Rv | None (WT) | 0.13 | 0.13 | 0.25 | <u><0.06</u> | <u><0.06</u> |
| M70 | FQ, STR, INH, RIF, and PZA | 0.25 | 1 | 0.25 | >16 | >16 |
| M28 | FQ, INH, RIF, EMB, and PZA | 0.25 | 2 | <u><0.06</u> | >16 | >16 |
| TN5904 | STR, INH, RIF, and PZA | 0.13 | 0.13 | 0.13 | >16 | 1 |

FQ, fluoroquinolone; STR, streptomycin; RIF, rifampicin; PZA, pyrazinamide; MOXI, moxifloxacin.

of >100 μM (Table 1). Similarly, the compounds were not toxic against the human macrophage cell line THP-1 ($IC_{50}$ > 50 μM). Both compounds had good potency against intracellular bacteria, with $IC_{50}$ and $IC_{90}$ in the sub-micromolar range (Table 1).

### Pharmacokinetic (PK) properties of AN12855

We selected the most potent diazaborine, AN12855, for in vivo murine PK analysis. AN12855 was formulated in 1% carboxymethyl cellulose, 0.1% Tween-80, and water adjusted to pH 6.2–6.5. In naive CD-1 mice, the PK of AN12855 is characterized by low clearance and moderate volume of distribution after i.v. and oral delivery with terminal elimination half-life of 3.5 h (Table S2). AN12855 had an acceptable oral bioavailability of 53% at 10 mg/kg, although total lung exposure to AN12855 was 33% lower than plasma area under curve (AUC) (Table S2). In protein binding studies, AN12855 was 88% bound in human serum and 98.5% bound in mouse plasma. The addition of 4% human serum albumin resulted in an eightfold reduction in potency

from 0.28 to 2.3 μM against *M. tuberculosis* Erdman TMCC 107. Thus, AN12855 has acceptable bioavailability but is highly protein bound.

### AN12855 is efficacious in an acute and chronic model of TB infection

Because AN12855 had acceptable PK properties, we selected this molecule for in vivo efficacy studies. In an acute mouse model of TB infection, AN12855 exhibited dose-dependent efficacy over a 9-d treatment regimen. In a parallel PK study, using naive C57BL/6 mice, there was a linear relationship between total drug exposure in plasma and drug dose as shown by $AUC_{0-last}$/dose (Table S2). Treatment with 10, 25, and 50 mg/kg resulted in 2.3, 2.7, and 3.7 $log_{10}$ reductions in *M. tuberculosis* lung burdens, respectively (Fig 4A and Table S3; $P$ < 0.05) and 2.1, 3.2, and 3.6 $log_{10}$ reduction in *M. tuberculosis* spleen burdens, respectively (Fig 4B and Table S3; $P$ < 0.05). The efficacy of AN12855 did not increase when dosed at 100 and 200 mg/kg (Fig S1A and B and Table S4). Dosing of AN12855 at 50

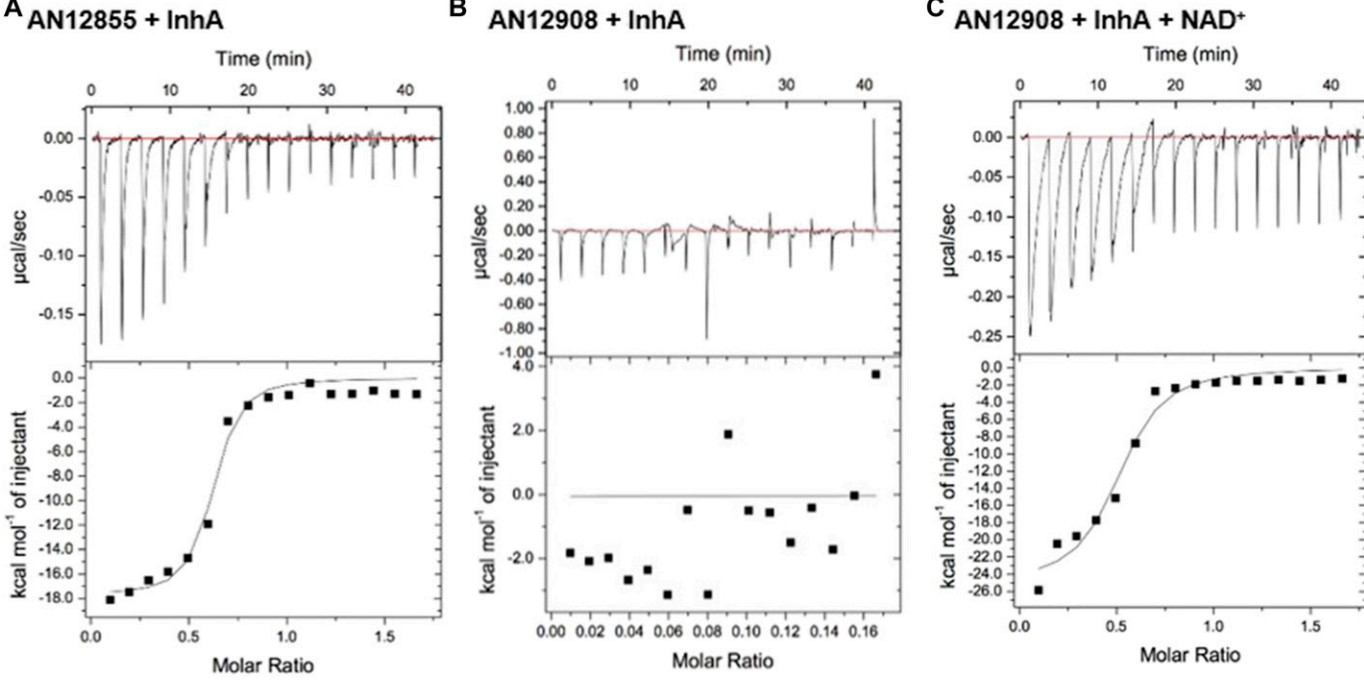

**Figure 2.  Thermodynamic analysis of the interaction between diazaborines and *M. tuberculosis* InhA.**
Binding of (A) AN12855 and (B, C) AN12908 with *M. tuberculosis* InhA as measured by ITC. For each compound, the interaction with InhA was measured in either (A, B) the absence of cofactor or (C) the presence of $NAD^+$. InhA, compounds, and cofactors were added as described in the Materials and Methods section.

**Table 5. Thermodynamic analysis of interactions between compounds and *M. tuberculosis* InhA.**

| Compound | Cofactor | $K_d$ (nM) | ΔH (cal/mol) | ΔS (cal/mol/deg) | N |
|----------|----------|-----------|--------------|------------------|---|
| AN12855 | None | 77 ± 31 | −17,740 ± 607 | −27 | 0.58 ± 0.014 |
| AN12908 | NAD⁺ | 847 ± 318 | −25,010 ± 1,496 | −56 | 0.49 ± 0.022 |

ΔH, change in enthalpy; ΔS, change in entropy; N, stoichiometry of binding.

mg/kg showed comparable efficacy to the frontline TB drug INH at 25 mg/kg (Fig 4A and B and Tables S3 and S4). In conclusion, AN12855 is efficacious in an acute model of TB infection. In a chronic BALB/c mouse model of TB infection, AN12855 dosed at 100 mg/kg showed similar efficacy to INH at 25 mg/kg and promoted reductions in lung burdens of 0.47, 0.81, and 1.73 $\log_{10}$ CFU by 2, 4, and 8 weeks of treatment, respectively (Fig 4C and Table S5; $P < 0.05$ at 2 and 8 weeks). Bacterial burdens in spleens of mice treated with INH and AN12855 (100 mg/kg) showed similar results as seen in lungs (Fig 4D and Table S5). AN12855 administered at 100 mg/kg showed similar efficacy as INH at 25 mg/kg with more than three $\log_{10}$ reductions in spleens after 8 weeks of treatment (Fig 4D and Table S5; $P < 0.05$). PK analysis of plasma collected at steady state (after 3 weeks of dosing) in the BALB/c efficacy study showed substantial drug levels in plasma of AN12855 (Table S2). The drug levels after long-term dosing were higher than those obtained in the earlier PK studies in uninfected CD1 mice following a single dose or C57BL/6 mice following short-term dosing. In conclusion, oral delivery of AN12855 has dose-dependent efficacy in both acute and chronic models of TB infection that are comparable with the frontline drug INH.

# Discussion

The high rate of resistance to INH, a key component of standard TB treatment, is a contributing factor in the emergence and spread of MDR-resistant strains of *M. tuberculosis*. Direct inhibitors of InhA are promising alternatives to INH as they inhibit a proven drug target of *M. tuberculosis*, have a lower rate of resistance, and are active against clinical isolates that are INH resistant. However, direct InhA inhibitors as a class of compounds are likely to have reduced potency against INH-resistant strains that have mutations in the *fabG1inhA* promoter, which overexpress InhA and should be taken into account when determining doses needed for strain coverage. In this study, we describe the identification of the diazaborines, a new class of antitubercular agents that directly inhibit InhA in *M. tuberculosis*. The lead compound, AN12855, has potent antitubercular activity in vitro, bactericidal activity against replicating bacteria, and there is a low frequency of resistance to AN12855. AN12855 is orally bioavailable and showed efficacy in both chronic and acute models of TB infection that was comparable with INH.

The co-crystal structure and ITC experiment results for AN12855 revealed a novel binding mode of the diazaborines that is independent of the cofactor NAD⁺. Although not a large molecule, the inhibitor AN12855 efficiently occupies both the cofactor and substrate-binding sites. The only other InhA inhibitor with a similar binding mode is the natural product pyridomycin (Hartkoorn et al, 2014). The crystal structure of AN12855 also allowed us to identify interactions with the catalytic residues Tyr158 and Lys165, and hydrophobic residues Ile215, Leu218, Met119, Ala157, and Phe149. These interactions are similar to other previously reported direct InhA inhibitors (Hartkoorn et al, 2014; Manjunatha et al, 2015; Martínez-Hoyos et al, 2016). With the exception of a recently reported triazole-based series (Spagnuolo et al, 2017), the interaction between the oxime group of AN12855 and Glu219 of InhA is generally not observed for direct InhA inhibitors. Only a small set of the strains with InhA mutations that conferred resistance to NITD-916 were cross-resistant to AN12855. These differences highlight the unique binding mode of AN12855 relative to other direct InhA inhibitors. The results of this current study suggest that the oxime group of AN12855 contributes to inhibitor potency by (i) increasing the binding affinity through a hydrogen-bonding interaction and the stabilization of the AN12855-InhA binary complex and (ii) lowering the lipophilicity of the AN12855 and improving *M. tuberculosis* cell permeability.

AN12855 is a promising lead compound for the development of novel TB therapeutics. AN12855 directly inhibits InhA, a proven drug target in *M. tuberculosis*, yet overcomes many of the problems associated with INH, including the high rate of resistance and the need for activation from a prodrug. AN12855 is different from other small molecular inhibitors of InhA in that it occupies both the cofactor and substrate-binding sites of InhA in a cofactor-independent manner. AN12855 had good efficacy in both the chronic and acute murine models of TB infection that was comparable with INH. The availability of structural and biochemical data will assist further development of these promising cofactor-independent inhibitors into a clinical candidate.

# Materials and Methods

### Chemical synthesis

AN2918 was synthesized according to patent US20070155699 A1 published on 5 July 2007. AN3438 was synthesized according to patent US8039450 B2 published on 18 October 2011. The syntheses of the other compounds used in this study are described in the Supplementary Information.

### InhA in vitro inhibition assay

Codon-optimized *M. tuberculosis* H37Rv His-InhA expressed from pET15b was purified from *E. coli* BL-21 using an Ni-NTA resin. InhA activity was monitored using previously defined protocols (Quemard et al, 1995; Parikh et al, 2000). Briefly, 30 mM PIPES (pH 6.8), 30 nM InhA, 0.25 mM NADH, 0.25 mM NAD⁺, and 150 mM NaCl were incubated for 30 min at room temperature with differing concentrations of compound of interest. An aliquot of 2-*trans*-dodecenoyl-CoA to make 0.3 mM was added to start the reaction and decreasing fluorescence was monitored over time using a Perkin–Elmer EnVision reader with excitation and emission wavelengths of

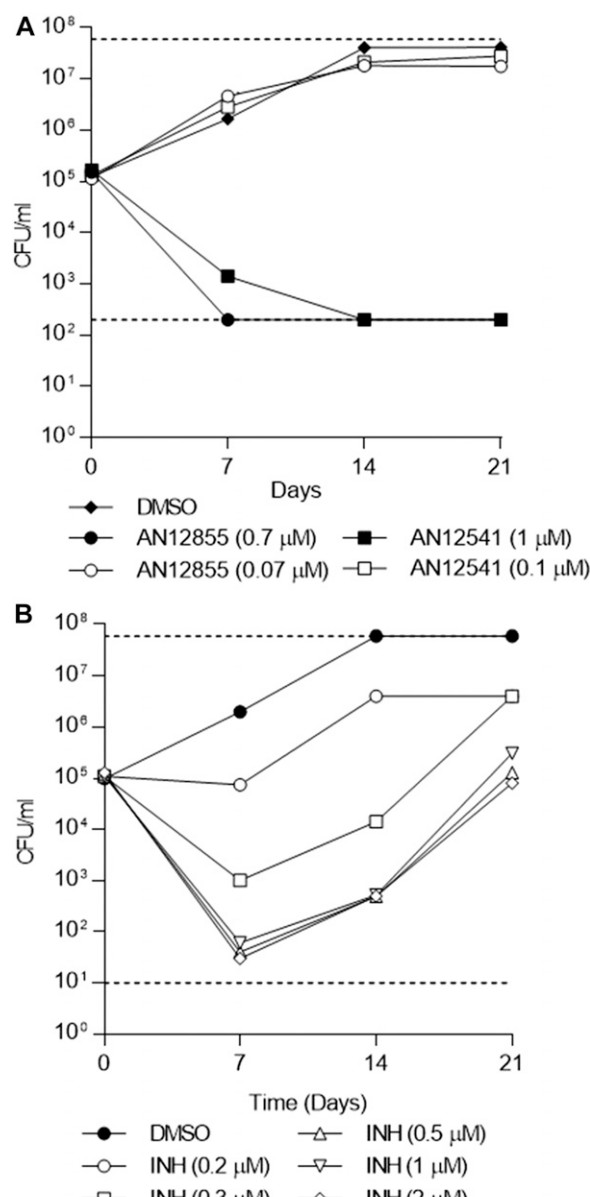

**Figure 3. In vitro kill kinetics of AN12541 and AN12855 against *M. tuberculosis.*** In vitro kill kinetics of AN12541 and AN12855 against *M. tuberculosis* under replicating conditions (A) AN12541 and AN12855 (B) INH. Limit of detection is marked by dashed lines.

340 and 445 nm. Active InhA converts fluorescent NADH into the less fluorescent NAD. The decrease of fluorescence over 60 min was monitored, and the $IC_{50}$ values were determined using four-parameter logistic equation implemented in the GraphPad Prism (GraphPad).

### Determination of compound $IC_{90}$ against drug-susceptible *M. tuberculosis*

The $IC_{90}$ of compound was determined as previously described (Ollinger et al, 2013). Briefly, bacterial growth was measured in the presence of test compounds. Compounds were prepared as 10-point twofold serial dilutions in DMSO and diluted into 7H9-Tw-

OADC medium in 96-well plates with a final DMSO concentration of 2%. Each plate included assay controls for background (medium/ DMSO only, no bacterial cells), zero growth (2 μM rifampicin), and maximum growth (DMSO only), as well as a rifampicin dose–response curve. Plates were inoculated with *M. tuberculosis* H37Rv (ATCC 25618) containing plasmid expressing the fluorescent protein DsRed (Zelmer et al, 2012). Growth was measured after 5 d by $OD_{590}$ and fluorescence (Ex 560/Em 590). Growth was calculated separately for $OD_{590}$ and relative fluorescent unit. Dose–response curves were generated using the Levenberg–Marquardt algorithm and the concentrations that resulted in 90% inhibition of growth were determined ($IC_{90}$).

### Determination of minimum inhibitory concentrations (MICs) against drug-resistant *M. tuberculosis*

MICs were determined against drug-resistant *M. tuberculosis* isolates TN5904, M70, and M28 in a 96-well microplate assay (Cheng et al, 2004; Gruppo et al, 2006; Palanisamy et al, 2008). Strains were cultured in 7H9 broth with 0.2% vol/vol glycerol and 10% vol/vol albumen, dextrose, and catalase to an $OD_{600}$ of 0.6–0.8. Suspensions were prepared to reach an inoculum of $10^5$ CFU per well in a total volume of 150 μl 7H9, whereas H37Rv was prepared to reach an inoculum of $5 × 10^4$ CFU per well. All compounds were dissolved in DMSO and prepared as twofold serial dilutions. The final concentration of DMSO was 2%. The plates were incubated at 37°C for 14 d and observed every 3–4 d to determine changes in growth. Growth of the bacteria was recorded by spectrophotometer readings at $OD_{600}$. An aliquot of Alamar Blue dye (15 μl) was added to each well at day 14. The Alamar Blue dye conversion was evaluated 48 h after addition (or day 7 for strain M28). H37Rv was used as a drug-susceptible control.

### Resistant mutant isolation and characterization

*M. tuberculosis* H37Rv-resistant mutants were selected on 7H10-OADC agar containing 5× or 10× solid media MIC for AN3438, AN6534, and AN12855 (Ioerger et al, 2013). Resistant mutants were confirmed by measuring the $IC_{90}$ in liquid medium (Ollinger et al, 2013) or the $MIC_{99}$ on solid medium (Sirgel et al, 2009). Mutations were identified by whole-genome sequencing (Ioerger et al, 2013) and confirmed by PCR amplification and sequencing of *inhA*.

### Kill kinetics

Kill kinetics of compounds were determined under replicating conditions using exponential-phase cultures of *M. tuberculosis* H37Rv ($5 × 10^5$ CFU/ml) in 7H9-OADC-Tw. CFUs were determined over 21 d by serial dilution and culture on 7H10-OADC plates for 3–4 weeks.

### Intracellular activity and THP-1 cytotoxicity

THP-1 cells (ATCC TIB-2202) were propagated in RPMI-1640, 10% vol/ vol FBS, 2 mM Corning glutagro (Corning), and 1 mM sodium pyruvate in a humidified atmosphere at 37°C, 5% $CO_2$. THP-1 cells were

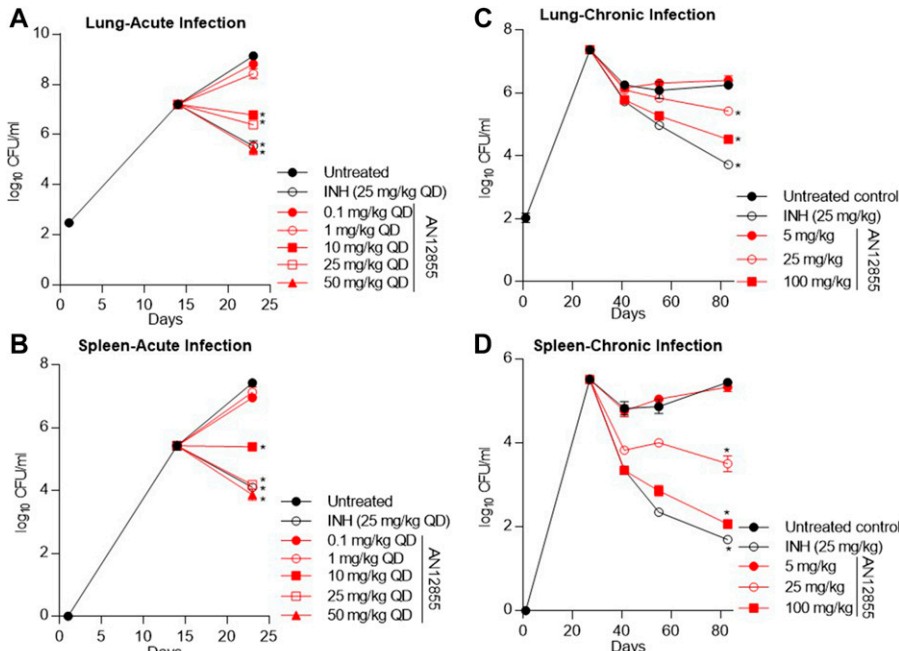

**Figure 4. AN12855 is efficacious in acute and chronic murine models of TB infection.**
**(A, B)** In vivo efficacy in a murine GKO (C57BL/6-Ifng^tm1Ts) model of acute TB. Compounds were dosed orally daily for 9 d after 14 d of infection (start) with a low-dose aerosol of *M. tuberculosis* Erdman. Mean (A) lung and (B) spleen $\log_{10}$ CFUs were determined from five mice at the start of treatment and 1 d following the last day of dosing. **(B, C)** In vivo efficacy in a murine BALB/c model of chronic TB infection. Compounds were dosed orally 5 d a week for 8 weeks after infection with *M. tuberculosis* Erdman with a low-dose aerosol 27 d prior (start). Mean (B) lung and (C) spleen $\log_{10}$ CFUs were determined from five to six mice at the start of treatment and following 2, 4, and 8 weeks of treatment. For (A–D), statistical analysis was performed as described in the Materials and Methods section; *$P < 0.05$.

differentiated into the macrophages using 80 nM phorbol myristate acetate overnight. The cells were infected overnight with *M. tuberculosis* (constitutively expressing LuxABCDE [Andreu et al, 2010]) at multiplicity of infection of 1. Infected THP-1 cells were harvested with Accutase (Innovative Cell Technologies), 5 mM EDTA solution, washed twice in PBS, and resuspended in fresh medium. Infected cells were seeded into 96-well plate at a concentration of $4 \times 10^4$ cells per well. Compounds were assayed for 72 h using a 10-point threefold serial dilution starting at 50 μM, and bacterial inhibition was assessed by relative light unit (RLU). Growth inhibition curves were fitted using the Levenberg–Marquardt algorithm. The $IC_{50}$ and $IC_{90}$ were defined as the compound concentrations that produce 50% or 90% of the intracellular growth inhibitory response, respectively. Uninfected THP-1 macrophages were propagated, harvested, and added to 96-well plates as described above. THP-1 viability was measured using CellTiter-Glo reagent (Promega) and reading RLU. Growth inhibition curves were fitted using the Levenberg–Marquardt algorithm. The $IC_{50}$ was defined as the compound concentration that reduced cell viability by 50%.

### HepG2 cytotoxicity

HepG2 human liver cells (ATCC HB 8065) were propagated in DMEM containing either 25 mM glucose or 10 mM galactose plus 10% vol/vol FBS, 1 mM sodium pyruvate, 2 mM Corning glutagro, 100 U/ml penicillin, and 100 μg/ml streptomycin. The cells were seeded in 384-well plates at 1,800 cells per well and incubated in a humidified incubator at 37°C, 5% $CO_2$. Compounds were solubilized in DMSO and assayed using a 10-point threefold serial dilution. Compounds were added 24 h post cell seeding to a final assay concentration of 1% DMSO and highest compound concentration of 100 μM. The cells were incubated for 72 h and viability measured using CellTiter-Glo

reagent (Promega) and reading RLU. Growth inhibition curves were fitted using the Levenberg–Marquardt algorithm. The $IC_{50}$ was defined as the compound concentration that produced 50% of the inhibitory response against HepG2 cells.

### ITC

ITC was performed using a GE MicroCal iTC$_{200}$. Purified *M. tuberculosis* InhA with the His tag removed and compounds of interest were mixed in 25 mM Hepes and 150 mM NaCl, pH 7.5, with a 1% or 3% DMSO solution. A total of 16 injections of 2 μl were performed. InhA solutions at 50 μM plus NADH or NAD at 750 μM in the calorimetric cell were titrated with AN3438 and AN12908 at 750 μM plus NADH or NAD at 750 μM. For InhA plus AN12855 interaction, InhA was used at 10 μM plus NADH or NAD at 600 μM in the calorimetric cell and titrated with AN12855 at 100 μM plus NADH or NAD at 750 μM. Compound solutions were incubated at 37°C during 1 h before titrations. The heat evolved after each ligand injection was obtained from the integral of the calorimetric signal. The resulting binding isotherms were analyzed by nonlinear least squares fitting of the experimental data to a single-site model. Analysis of the data was performed by using MicroCal Origin software (OriginLab version 7 [OriginLab]). The experiments were performed at least twice. The variability in the binding experiments was estimated to be 5% for binding enthalpy and 10% for both the binding affinity and the number of sites.

### Crystallization, structure determination, and refinement

Crystals of InhA with the His tag removed in 20 mM PIPES, pH 7.3, and 50 mM NaCl were grown in the presence of either (i) 3.5 mM NAD$^+$ and 420 μM AN2918 in 0.1 M ADA/NaOH, pH 6.8, 12.0% wt/vol polyethylene glycol (PEG) 4000, 0.25 M ammonium acetate, and

cryoprotected in 40% MPD supplemented with 0.1 mM compound; (ii) 3.5 mM NAD$^+$ and 420 μM AN3438 in 0.1 M ADA/NaOH, pH 6.8, 1.0% wt/vol DMSO, 14.0% wt/vol PEG 4000, 0.25 M ammonium acetate, and cryoprotected in 40% MPD supplemented with 0.1 mM compound; or (iii) 880 μM AN12855 grown in the Morpheus_d6 focus screen, condition d6: 9%–11% PEG 8000, 18%–22% ethylene glycol, 100 mM Hepes, pH 6.5–8.5, 18–22 mM each of 1,6-hexanediol, 1-butanol, 1,2-propanediol, 2-propanol, 1,4-butanediol, 1.3-propanediol, and directly cryoprotected in the well solution supplemented with 0.1 mM compound. All diffraction data sets were collected at the synchrotron APS beamline 21-IDF. The structures were solved by molecular replacement with 1ENY using the CCP4 program Phaser and Refmac (CCP4). For InhA-AN2918, four copies of InhA were placed per asymmetric unit. For InhA-AN3438, six copies of InhA were placed per asymmetric unit. For InhA-AN12855, 1 copy of InhA was placed per asymmetric unit. Refinement was performed by iterative cycles of model building in Coot64 (0.7-prei) and refinement in Refmac (CCP4). Refinement statistics were included in the Supplementary Information. Atomic coordinates have been deposited at RCSB protein data bank under accession codes 5VRN (AN3438), 5VRM (AN2918), and 5VRL (AN12855).

### Murine PK analysis

Murine PK studies of AN12855 were conducted by using female CD-1, C57BL/6, and infected BALB/c mice. Mice received the test article by either i.v. tail vein injection or oral (p.o.) gavage. Naive CD-1 mice were administered a single dose of the test article, and blood samples were collected via cardiac puncture at specific time points through 24 h (K2EDTA as an anticoagulant) and processed for plasma. Lung tissue was processed by homogenizing 0.1 g of tissue with 0.3 ml of 5 mM ammonium acetate. C57BL/6 mice were dosed once daily for 4 d with samples collected on day 5. Plasma samples from C57BL/6 mice were collected at 0.5, 2, 5, and 8 h post dosing. C57BL/6 mice were euthanized at only two time points (5 and 8 h post dosing). Because of this, the lung $C_{max}$ for C57BL/6 is likely underestimated. For BALB/c PK analysis, plasma samples were collected via submandibular bleeds at steady state from infected mice from the efficacy study. Antibiotic concentrations in the plasma samples were analyzed by liquid chromatography–tandem mass spectrometry using an API4000 QTRAP instrument (AB Sciex). Quantification was achieved by comparing the analyte/internal standard peak areas with the internal standard AN3365 (Hernandez et al, 2013). The limit of quantitation was 1 or 2 ng/ml. PK analyses of the mean concentration–time profiles were performed by using WinNonlin Pro version 5.2. Protein binding studies in mouse plasma and human sera were performed as described (Beer et al, 2009). The determination of compound MIC in the presence of 4% human serum albumin was performed as described (Beer et al, 2009).

### Ethics statement

The animal protocols involving mice were approved by Colorado State University's Institutional Animal Care and Use Committee. Mice were housed in a biosafety level III animal facility and maintained with sterile bedding, water, and mouse chow.

### Murine model of acute TB infection

8- to 10-week-old female specific pathogen–free C57BL/6-Ifng$^{tm1Ts}$ mice (interferon gamma receptor knockout mice [GKO]) were purchased from Jackson Laboratories. The mice were infected with *M. tuberculosis* Erdman (TMCC 107) via a low-dose aerosol exposure in a Glas-Col aerosol generation device (Glas-Col Inc.) as described previously (Lenaerts et al, 2005). At 1 d post-aerosol, three mice were sacrificed to verify the uptake of ~100 CFU of *M. tuberculosis* Erdman per mouse. Each treatment group consisted of five mice, and treatment was started at 14 d post-aerosol infection and continued for 9 consecutive days. Five infected mice were sacrificed at the start of treatment as pretreatment controls. Drugs were administered daily by oral gavage in a volume of 200 μl per mouse. For endpoint analysis, the mice were euthanized one day following the end of treatment, and the lungs and spleens were collected. The left lung lobe or whole spleens were homogenized for enumeration of CFU by plating dilutions of the organ homogenates on Middlebrook 7H11 medium supplemented 10% vol/vol OADC, 0.03 mg/ml cycloheximide, and 0.05 mg/ml carbenicillin. The data were expressed as mean log10 CFU ± the SEM for each group.

### Murine model of chronic TB infection

6- to 8-week-old female specific pathogen–free immunocompetent BALB/c mice (Charles River) were infected with *M. tuberculosis* Erdman (TMCC 107) via low-dose aerosol exposure as described previously (Lenaerts et al, 2005). At 1 d post-aerosol, three mice from each run were sacrificed to verify the uptake of ~100 CFU of bacteria per mouse. Each group consisted of five to six mice at each time point. Treatment was started at 4 weeks post-aerosol infection and continued for 8 weeks. Five infected mice were sacrificed at the start of treatment as pretreatment controls. Drugs were administered daily (5 d per week) by oral gavage for 8 weeks in a volume of 200 μl per mouse. After 2, 4, and 8 weeks of treatment, five to six mice from each group were sacrificed. For endpoint analysis, mice were euthanized 3 d following the last administered drug dose, and the lungs and spleens were collected. The left lung lobe or whole spleens were homogenized for enumeration of CFU. The data were expressed as mean log10 CFU ± the SEM for each group.

### In vivo efficacy statistical analysis

Data were evaluated by a one-way analysis of variance followed by a multiple comparison analysis of variance by a one-way Tukey test (SAS Software program). Differences were considered significant at the 95% level of confidence.

### Data deposition

Atomic coordinates have been deposited at RCSB protein data bank under accession codes 5VRL (InhA+AN12855), 5VRM (InhA+AN2918), and 5VRN (InhA+AN3438).

# Supplementary Information

# Acknowledgements

We thank James Ahn, Torey Alling, Olena Anoshchenko, Mai Bailey, John Evre, Megan Files, Megha Gupta, Douglas Joerss, Juliane Ollinger, Yulia Ovechkina, Anisa Tracy, and Dean Thompson for technical assistance. This research was supported with funding from the Bill & Melinda Gates Foundation and with National Institutes of Health grant GM102864 to PJ Tonge.

## Author Contributions

Y Xia: conceptualization, data curation, formal analysis, investigation, methodology, and writing—review and editing.

Y Zhou: conceptualization, data curation, formal analysis, investigation, methodology, and writing—review and editing.

DS Carter: conceptualization, data curation, formal analysis, investigation, methodology, and writing—review and editing.

MB McNeil: conceptualization, data curation, formal analysis, investigation, methodology, and writing—original draft, review, and editing.

W Choi: conceptualization, data curation, formal analysis, investigation, methodology, and writing—review and editing.

J Halladay: conceptualization, data curation, formal analysis, investigation, methodology, and writing—review and editing.

P Berry: conceptualization, data curation, formal analysis, investigation, and writing—review and editing.

W Mao: conceptualization, data curation, formal analysis, investigation, methodology, and writing—review and editing.

V Hernandez: conceptualization, data curation, formal analysis, investigation, methodology, and writing—review and editing.

T O'Malley: conceptualization, data curation, formal analysis, investigation, methodology, and writing—review and editing.

A Korkegian: conceptualization, data curation, formal analysis, investigation, and writing—review and editing.

B Sunde: conceptualization, investigation, methodology, and writing—review and editing.

L Flint: conceptualization, formal analysis, investigation, methodology, and writing—review and editing.

LK Woolhiser: conceptualization, data curation, formal analysis, investigation, and writing—review and editing.

MS Scherman: conceptualization, data curation, funding acquisition, investigation, methodology, and writing—review and editing.

V Gruppo: conceptualization, data curation, formal analysis, investigation, and writing—review and editing.

C Hastings: conceptualization, data curation, formal analysis, investigation, and writing—review and editing.

GT Robertson: conceptualization, data curation, formal analysis, investigation, and writing—review and editing.

TR Ioerger: conceptualization, investigation, methodology, and writing—review and editing.

JC Sacchettini: investigation, methodology, and writing—review and editing.

PJ Tonge: investigation, methodology, and writing—review and editing.

A Lenaerts: conceptualization, data curation, formal analysis, supervision, investigation, methodology, and writing—review and editing.

T Parish: conceptualization, data curation, formal analysis, supervision, funding acquisition, investigation, and writing—review and editing.

MR Alley: conceptualization, data curation, formal analysis, supervision, investigation, methodology, and writing—review and editing.

## Conflict of Interest Statement

The authors declare that they have no conflict of interest.

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
