## [Reviewer comments · Life Science Alliance]

Discovery of a cofactor-independent inhibitor of *Mycobacterium tuberculosis* InhA

Yi Xia, Yasheen Zhou, David S. Carter, Matthew B. McNeil, Wai Choi, Jason Halladay, Pamela W. Berry, Weimin Mao, Vincent Hernandez, Theresa O'Malley, Aaron Korkegian, Bjorn Sunde, Lindsay Flint, Lisa K. Woolhiser, Michael S. Scherman, Veronica Gruppo, Courtney Hastings, Gregory T. Robertson, Thomas R. Ioerger, Jim Sacchettini, Peter J. Tonge, Anne J. Lenaerts, Tanya Parish, M. R. K. Alley

DOI: 10.26508/lsa.201800025

Review timeline:	Submission Date:	25 January 2018
	1 st Revision Received:	25 January 2018
	1 st Editorial Decision:	29 January 2018
	2 nd Revision Received:	16 May 2018
	2 nd Editorial Decision:	18 May 2018
	3 rd Revision Received:	18 May 2018
	Accepted:	22 May 2018

Report:

(Note: Letters and reports are not edited. The original formatting of letters and referee reports may not be reflected in this compilation.)

1st Revision – authors' response

25 January 2018

Referee #1 (Remarks for Author):

Multidrug (MDR) and extensively drug (XDR) resistant strains of *Mycobacterium tuberculosis* are a major public health threat. Building upon previous efforts, the authors report on the identification of diazaborines as a new class of InhA inhibitors. Following identification of three initial hits in compound screening and further chemical synthesis, the authors identify lead compound AN12855 and subsequently use a variety of techniques to further characterize AN12855, including structural analysis and in-vivo efficacy studies. This is a solid piece of work, although the efforts on chemical synthesis are somewhat limited and the microbiological investigations in part lack depth.

1. Major

1.1 In addition to mutational KatG alterations, mutations in the *inhA* gene are a major source of INH resistance in *M. tuberculosis*. The diazaborines should be tested against clinical isoniazid resistant *M. tuberculosis* strains carrying the following INH resistance mutations:

- mutations in *inhA* promoter
- mutations in *inhA* structural gene
- combined mutations in *inhA* promoter and structural gene

These experiments will allow to address the critical question of whether diazaborines as a new class of InhA inhibitors are also effective against this important group of IHN resistant *M. tuberculosis* isolates.

We have included additional experimental work in Table 3. We tested AN12855, against a collection of 26 M. tuberculosis strains with mutations in the fabG1inhA promoter and/or inhA coding sequence. These isogenic mutant strains are described in McNeil, MB et al. 2017. Tuberculosis. <https://doi.org/10.1016/j.tube.2017.09.003>. Several of these strains have mutations associated with INH resistance in clinical isolates, including the fabG1inhA (c-15t) promoter mutant and InhA(S94A). These two mutations represent the majority of non-KatG mediated resistance to INH in clinical isolates.

1.2 Table 1 summarizes the compounds synthesized (n = 6). Are these all compounds which have been synthesized? More efforts in compound optimization would be appreciated.

We agree that a full SAR effort on the series would be of interest, but this was not the focus of the current paper.

1.3 What is the protein binding of AN12855? This is pretty relevant, as some of the new anti TB compounds are compromised by excessive protein binding, e.g. bedaquiline and delamanid.

We have included additional experimental work and data. AN12855 was 98.5% protein bound in mouse and 88% bound in human. Protein binding reduced efficacy in the serum shift assay as expected.

1.4 It is surprising that there is no dose-dependency in the killing assay for non-replicating M. tuberculosis (Fig. 3B). Both doses of compounds, i.e., 0.7 and 0.07 μ M for AN12855 and 1.0 and 0.1 μ M for AN12541, show essentially the same killing in this assay. This is somewhat difficult to understand - at least for this reviewer.

Compounds can have either time-dependent or concentration-dependent bactericidal activity – for example, see refs (i) Bakker-Woudenberg IAJM et al (2005) Antimycobacterial agents differ with respect to their bacteriostatic versus bactericidal activities in relation to time of exposure, mycobacterial growth phase, and their use in combination. Antimicrobial Agents and Chemotherapy 49: 2387-2398, (ii) de Steenwinkel JEM, et al IAJM (2010) Time-kill kinetics of anti-tuberculosis drugs, and emergence of resistance, in relation to metabolic activity of Mycobacterium tuberculosis. Journal of Antimicrobial Chemotherapy 65: 2582-2589 and (iii) Vogelman B and Craig WA (1986) Kinetics of antimicrobial activity. Journal of Pediatrics 108: 835-840. The activity of the compounds is time-dependent as defined by the Clinical Laboratory Standards Institute (Approved Guidelines (M26-A). We have clarified this in the text. This is not unusual.

2. Minor

2.1 Introduction should include mutations in inhA promoter and structural gene as one of the major INH resistance mechanisms.

This has been included in the introduction of the revised manuscript

2.2 Introduction, lines 27 to 29: "A promising observation from these studies is the lower frequency of resistance for direct inhibitors of InhA with 1×10^{-8} for NITD-916 and GSK626 compared with 1×10^{-5} for INH." This is not a fair statement. It is well established that INH resistance mutations generated in-vitro have little, if anything, in common with the INH resistance mutations observed in clinical drug resistant isolates (Bergval et al. J. Antimicrob. Chemother. 2009, 64: 515). Thus, for INH no conclusions from the in-vitro frequency of resistance to the clinical situation are possible.

We agree that INH mutants raised in vitro generally have little correlation to mutations observed in clinical isolates. However, differences in the rate of resistance between INH and direct InhA inhibitors observed in vitro is of significance. We have added the following statement to the introduction "Further studies are required to determine if differences in in vitro frequencies of resistances correlate with in vivo resistance frequency"

2.3 Provide the resistance mutations for the clinical drug resistant strains shown in Table 3.

The resistance mutations in these clinical isolates are not known.

2.4 Author contributions are incomplete, e.g. PJT, JS, and TRI are missing.

This has been corrected.

Referee #2 (Comments on Novelty/Model System for Author):

Values per mice, means, standard deviation and statistics of the in vivo efficacy experiment are not provided

We have included additional information as requested by the reviewer.

Fig 3 should also be completed with error bars.

This experiment was performed once.

Material and methods need to be completed with some important lacking information (including the nature of the enzymatic substrate)

Information has been included.

Referee #2 (Remarks for Author):

Discovery of a cofactor-independent inhibitor of Mycobacterium tuberculosis InhA

This document describes the identification of diazaborines as a new class of direct inhibitors of inhA. This approach has been followed with variable success by different groups and has produced many reports in the last 5 to 10 years. The first part of the manuscript gives little novelty, with results partially expected as benzodiazaborines where indeed reported actives against FabI of E.coli. The most interesting aspect of the present report is the identification of a NAD-independent family of compounds showing very strong activity in vitro, and good activity in vivo, including in a non-replicating in vitro model of TB.

Whereas globally convincing, some improvements of the manuscript are required and some additional experiments are needed to confirm InhA as the primary and lethal target of these compounds.

Authors claims that AN compounds are active against INH resistant clinical strains.

What is the efficacy of the "AN compounds" on strains mutated in the inhA promoter, in particular in strains mutated in C-15T promoter region of the mabA-inhA operon ?

It would be expected that these C-15T strains show some level of resistance to the "ANcompounds", which would reinforce InhA as the target of these compounds. But at the same time, if it is the case, authors should be more cautious when writing "active against INH resistant isolates" and restrict it to KatG mutants.

For resistance profiles against a variety of inhA coding and promoter mutants see response to reviewer one.

In the same line, status of the KatG and InhA genes (including promoter) of the strains described in Table 3 should be added.

In particular, what is the InhA status of strain TN5904 ? The low level of resistance to INH suggest a C_15T or an "inhA-ORF" type of mutation. If it is one of these 2 cases, the absence of resistance to AN12855 is unexpected and thus would need to be discussed.

See response to similar question for reviewer one

Authors should analyze by thin-layer chromatography mycolic acid biosynthesis in M. tuberculosis treated with diazaborines, in comparison to INH treatment.

Investigating the effects of these compounds on mycolic acid biosynthesis, although interesting, we believe is beyond the scope of this current manuscript.

Activity of diazaborines should be measured on a strain overexpressing InhA.

We have tested the activity against an isogenic strain with a mutation in the promoter which leads to over-expression. These data are included in the revised manuscript.

It would be interesting to test the cross-resistance of the diazaborine resistant mutants to triclosan. *In this current study only InhA (D148E, M161L, R195G, I215S) were resistant to the diazaborine AN12855. None of these mutants were cross resistant to triclosan (McNeil et al. 2017. Tuberculosis. In press. <https://doi.org/10.1016/j.tube.2017.09.003>). We have included the reference.*

It is very exciting to see the activity of the compounds against replicating and non-replicating *M. tuberculosis*. At the same time, it is quite surprising to see that compounds has a much better activity against non-replicating bacteria than against replicating ones, which is not expected considering InhA as the target. As a control, it is important to show how behaves INH in this non-replicative assay.

The very low active dose on non-replicative bacteria may suggest that the target could be different under the two growth conditions. If InhA is still the target, mutation C-15T or plasmid overexpressing InhA should logically influence the MIC on this non-replicating assay. This should be tested.

We have included data for INH for comparison. We see that INH also has activity against non-replicating organisms in the starvation model. To the best of our knowledge this is the first evidence demonstrating that targeting InhA is bactericidal against nutrient starved non replicating M. tuberculosis. This data has been included in the revised manuscript.

Values per mice, means, standard deviation and statistics of the in vivo efficacy experiment should be summarized in a supplementary table.

Included

Fig 3 should also be completed with error bars.

See response to similar question above

Minor additional comments:

The most interesting compounds of the series is AN12855. Surprisingly, the remarkable IC90 of this compounds (0.05 uM) is not reported in the main text, but only appears in table 2.

This data has been included in the revised manuscript.

In table 2: I suppose that all IC90 (table 2) were determined on solid medium, and not only the IC90 of AN3438. If this is the case, move the "A" to the cell "IC90(uM)". If not, it is not recommended to compare MIC on solid for some compounds versus MIC liquid for others.

We have made changes to Table 2 to make it clear as to the assay conditions of each experiment.

The MIC of isolated resistant mutants against of AN3438 was determined on solid media, whilst the MIC against AN6534 and AN12855 was determined in liquid media. The purpose of these experiments was to compare the resistance profiles of isolated resistant mutants to the parental H37Rv. Consequently, the method of determining resistance does not mask the result that the isolated mutants have differing resistance profiles against diazaborines. These changes have been included in the revised manuscript.

Lines 127 to 132 describe correctly figures 3A and 3B. However, I don't think authors should use the wording "concentration-dependent" or "time-dependent" to summarize what is observed. These terminologies are mainly used to describe the pharmacokinetics compartment of drugs, as they define the in vivo properties of compounds based on their "peak over MIC" or "time over MIC", but not their properties in vitro.

The definitions of concentration or time dependent killing used in this study to describe in vitro kill kinetics are consistent with previously published data and approved guidelines provided by the Clinical Laboratory Standards Institute (Approved Guidelines (M26-A)) (Bakker-Woudenberg, Van Vianen et al., 2005, CLSI, 1999, de Steenwinkel, de Knegt et al., 2010, Vogelmann & Craig, 1986).

For uniformity of the results: The antibacterial activity of compounds is sometimes reported as IC50 (line 138, table 1) and sometimes IC90 (line 46, 51, 89, 260). Using IC90 in both cases would allow a clear comparison, especially in Table 1.

In this study the IC₅₀ is reported when referring to the activity against intracellular M. tuberculosis within THP-1 cells or for activity against purified InhA. The IC₉₀ is reported when referring to activity against extracellular M. tuberculosis. As per the reviewers request we have included the IC₉₀ of AN12855 and AN12541 against intracellular M. tuberculosis in Table 1 of the revised manuscript.

Line 286. It is not clear whether "cell viability" refers to macrophage viability (toxicity assay) or to bacteria viability.

Does the Lux assay (RLU) measure "bacterial growth inhibition" (as written on line 281), or bacteria viability (as written on line 280) ?

We have clarified this in the revised manuscript.

Description of the InhA in vitro inhibition assay is incomplete. What is the lipid attached to CoA in your assay ?

This information has been included. See response to similar question above.

Line 344: it is reported that C57BL/6 mice were dosed 5h and 8h post dosing. In this case, what was the model used to calculate the T_{max} value (0.5 h, reported in Supplemental Table 2) ?

T_{max} in the lung homogenates of C57BL/6 mice was calculated as 5 hours post dosing. Of the two available time points (i.e. 5 and 8 hours post dosing), this was the time point that provided the highest concentration.

As noted by the reviewer the T_{max} in C57BL/6 plasma was calculated as 0.5 hours post dosing. Of the available time points, this was the time point that provided the highest concentration. Plasma samples from C57BL/6 mice were collected at 0.5, 2, 5 and 8 hours post dosing. We have corrected the revised manuscript to highlight that there were time differences in plasma and lung sample collection.

Line 45: correct AN6354 AN6534

Line 46: change 35 μ M by 36 μ M for consistency with table 1

Line 48: correct AN3848 AN3438

Line 49: change > by (greater than or equal to)

Line 50: Change "Five of the strains" by "Four of the strains" and change > by (greater than or equal to)

Line 74: benzodiazaborine instead of "benzodiazoborine"

Verify presence of space between numbers and Unit, in particular in "Materials and Methods"

Fig.4 A: replace "025 mg/kg" by "25 mg/kg"

Line 261: delete "either"

Line 306: delete "mM"

Line 302: delete "in" mixed

Line 317: delete "R5645"

Line 318: delete the second "grown" of the line

Line 321: "containing grown" ?

All the above changes have been made.

1st Editorial Decision

29 January 2018

Thank you for transferring your manuscript entitled "Discovery of a cofactor-independent inhibitor of Mycobacterium tuberculosis InhA" to Life Science Alliance.

The manuscript was assessed by expert reviewers at another journal twice, and based on this prior assessment we would like to invite you to submit a revision addressing the reviewers' remaining key concerns, as outlined here.

We would like to ask you to address the remaining concerns by down-toning your statements, and by following reviewer #1's advice to exclude the killing assay data for non-replicating M. tuberculosis, and the corresponding claims on bactericidal activity against non-replicating bacilli. Furthermore, please address this referee's points 2 and 3 and mention the limited bioavailability of the drug in the manuscript.

Thank you for this interesting contribution to Life Science Alliance. We are looking forward to receiving your revised manuscript.

REFeree REPORTS

Referee #1 (Remarks for Author):

In the revised version the authors address some, but not all of the reviewers' concerns.

Three major issues of criticism remain unresolved:

1. Both referees have problems with the killing assay for non-replicating *M. tuberculosis*. In the view of this reviewer, the authors' response does not adequately address this concern. A further limitation is that this experiment was only performed once. Detailed dose-response curves, repeat testing, and testing of strains with mutation *InhA C-15T* is requested. An alternative option would be to eliminate these data, i.e., the killing assay for non-replicating *M. tuberculosis*, and the corresponding claims on bactericidal activity against non-replicating bacilli from the paper - this decision is up to the editor.
2. The authors should be more cautious when writing "active against INH resistant isolates". It should be made clear already in the abstract that AN compounds are not active against INH resistant isolates with mutation in *inhA* - this is quite a significant limitation which should clearly be spelled out in the abstract as well as in the discussion.
3. The resistance mutations for the clinical drug-resistant isolates (former Table 3, now Table 4) need to be provided - at least for INH. This simply is state-of-the-art. The authors' response that the resistance mutations are not known is not acceptable.

In addition, I am concerned about the protein binding of AN12855 which was not specified in the previous version of the manuscript and has been determined upon request of this referee. A protein binding of 88% to 98% is pretty high and points to possible limitations in bioavailability. Thus, the seemingly very strong activity in-vitro is somewhat compromised - in-vivo in mice AN12855 is 4-fold less potent than isoniazid (100 mg/kg AN12855 are required in lung and spleen chronic infection to give similar activity as 25 mg/kg isoniazid).

Referee #2 (Remarks for Author):

Authors responded to most of the remarks of the reviewers.

Additional remarks:

It should be clearly indicate in the text or in the legend of the corresponding table which one of the strains listed in table 3 is the "isogenic strain with a mutation in the promoter which leads to (*inhA*) over-expression".

The increase of resistance to AN12855 of bacteria over-expressing *InhA* is a good indication suggesting *InhA* as the target of this compound. However, such mutation also produces the over-expression of *MabA*. I believe that comparing the impact of AN12855 to the one of INH onto the synthesis of mycolic acid would have substantially reinforced the understanding of the mode of action of this new compound.

Regarding the table of statistics of the mice experiments, it should be indicated that the " * " refers to the comparison between the various treatment conditions and the negative control.
To be accepted for publication, authors should significantly improve the English of the manuscript:

See few examples of sentences picked specifically in the "discussion" section (many others remain in the rest of the text):

223. The lead compound, AN12855, has potent anti-tubercular activity in vitro, bactericidal activity against replicating and nutrient starved non-replicating bacteria, and (produces) a low frequency of resistance.

232. with this binding mode is a (the) natural product pyridomycin

233. The crystal structure of (the complex InhA-AN12855) also identified interactions...

Please read carefully lines 246 to 249 to avoid « statements of the obvious ».

In these sentence, the mix of « dormant TB, persisters and latent TB » is difficult to follow.

249. Whereas the starvation model used in this work mimics what is happening in humans remains to be demonstrated. Thus, remove the word « these » in line 249 to avoid direct comparison between the non-replicating model used in this work and the physiological state of *M. tuberculosis* in human disease.

260. AN12855 is unique (...) to other small molecular inhibitors of InhA in that (it) occupies...

262. AN12855 (??? showed ???) good efficacy in both the chronic and acute murine models of TB infection that were (was) comparable...

2nd Revision Received

16 May 2018

Response to reviewers

*We would like to ask you to address the remaining concerns by down-toning your statements, and by following reviewer #1's advice to exclude the killing assay data for non-replicating *M. tuberculosis*, and the corresponding claims on bactericidal activity against non-replicating bacilli. Furthermore, please address this referee's points 2 and 3 and mention the limited bioavailability of the drug in the manuscript.*

We have addressed all points as requested and detailed below.

*1. Both referees have problems with the killing assay for non-replicating *M. tuberculosis*. In the view of this reviewer, the authors' response does not adequately address this concern. A further limitation is that this experiment was only performed once. Detailed dose-response curves, repeat testing, and testing of strains with mutation *InhA C-15T* is requested. An alternative option would be to eliminate these data, i.e., the killing assay for non-replicating *M. tuberculosis*, and the corresponding claims on bactericidal activity against non-replicating bacilli from the paper - this decision is up to the editor.*

We have removed the data on non-replicating bacteria from the figures and text.

*2. The authors should be more cautious when writing "active against INH resistant isolates". It should be made clear already in the abstract that AN compounds are not active against INH resistant isolates with mutation in *inhA* - this is quite a significant limitation which should clearly be spelled out in the abstract as well as in the discussion.*

*We feel that we have already addressed these points in the manuscript. We did not mention INH-resistant isolates in the abstract; we wrote they were active against "drug resistant clinical isolates". We have modified to read "several drug resistant clinical isolates". We also addressed this point in the discussion. We previously included the sentence "However, direct *InhA* inhibitors as a class of compounds are likely to have reduced potency against INH resistant strains that have mutations in the *fabG*/*inhA* promoter, which overexpress *InhA*, and should be taken into account when determining doses needed for strain coverage." We think that this already addressed the reviewer's comment.*

3. The resistance mutations for the clinical drug-resistant isolates (former Table 3, now Table 4) need to be provided - at least for INH. This simply is state-of-the-art. The authors' response that the resistance mutations are not known is not acceptable.

We have provided this information in the text and in the supplementary data.

In addition, I am concerned about the protein binding of AN12855 which was not specified in the previous version of the manuscript and has been determined upon request of this referee. A protein binding of 88% to 98% is pretty high and points to possible limitations in bioavailability. Thus, the seemingly very strong activity in-vitro is somewhat compromised - in-vivo in mice AN12855 is 4-fold less potent than isoniazid (100 mg/kg AN12855 are required in lung and spleen chronic infection to give similar activity as 25 mg/kg isoniazid).

We measured oral bioavailability for AN28155 at 53%. We stated that "AN12855 has acceptable bioavailability but is highly protein bound" in line 178. Protein binding was lower in human plasma (88%) than mouse plasma (98%).

It should be clearly indicate in the text or in the legend of the corresponding table which one of the strains listed in table 3 is the "isogenic strain with a mutation in the promoter which leads to (inhA) over-expression".

We provided this information already in both the text and in the Table. Table 3 shows the mutation in the promoter region in the first column. It is denoted under the column "fabGIinhA promoter" and the nucleotide number is given. The text says "The fabGIinhA c-15t promoter mutant strain that over expresses InhA demonstrated a five-fold increase in IC₉₀ against AN12855 (Table 3)." In lines 110-112.

The increase of resistance to AN12855 of bacteria over-expressing InhA is a good indication suggesting InhA as the target of this compound. However, such mutation also produces the over-expression of MabA. I believe that comparing the impact of AN12855 to the one of INH onto the synthesis of mycolic acid would have substantially reinforced the understanding of the mode of action of this new compound.

As noted, we feel these studies would not add to the manuscript. Mycolic acid synthesis studies would not distinguish between inhibition of synthesis or export. In fact inhibition of MabA would also lead to reduced synthesis of mycolic acids. The inclusion of co-crystals of inhibitors with InhA, direct evidence of inhibition of InhA enzyme activity, and the fact that mutations in inhA were the only ones seen is stronger evidence.

*Regarding the table of statistics of the mice experiments, it should be indicated that the " * " refers to the comparison between the various treatment conditions and the negative control.*

Added

To be accepted for publication, authors should significantly improve the English of the manuscript: See few examples of sentences picked specifically in the "discussion" section (many others remain in the rest of the text):

We have made the changes suggested below.

223. The lead compound, AN12855, has potent anti-tubercular activity in vitro, bactericidal activity against replicating and nutrient starved non-replicating bacteria, and (produces) a low frequency of resistance.

232. with this binding mode is a (the) natural product pyridomycin

233. The crystal structure of (the complex InhA-AN12855) also identified interactions...

Please read carefully lines 246 to 249 to avoid « statements of the obvious ».

In these sentence, the mix of « dormant TB, persisters and latent TB » is difficult to follow.

Lines deleted

249. Whereas the starvation model used in this work mimics what is happening in humans remains to be demonstrated. Thus, remove the word « these » in line 249 to avoid direct comparison between the non-replicating model used in this work and the physiological state of M. tuberculosis in human disease.

Experiments deleted

We have made the changes suggested below.

260. AN12855 is unique (...) to other small molecular inhibitors of InhA in that (it) occupies...

262. AN12855 (??? showed ???) good efficacy in both the chronic and acute murine models of TB infection that were (was) comparable...

Thank you for submitting your revised manuscript entitled "Discovery of a cofactor-independent inhibitor of Mycobacterium tuberculosis InhA".

I appreciate the introduced changes, and I am happy to accept your manuscript in principle for publication in Life Science Alliance. Congratulations on this very nice work!